

# The oral microbiome of patients with axial spondyloarthritis compared to healthy individuals

Jordan E. Bisanz[1,2], Praema Suppiah[3], W. Murray Thomson[3], Trudy Milne[3,4], Nigel Yeoh[5], Anita Nolan[5,6], Grace Ettinger[1,2], Gregor Reid[1,2,7], Gregory B. Gloor[2,8], Jeremy P. Burton[1,2,7], Mary P. Cullinan[3,4] and Simon M. Stebbings[5]

[1] Department of Microbiology and Immunology, Western University, London, Ontario, Canada
[2] Canadian Centre for Human Microbiome and Probiotic Research, Lawson Health Research Institute, London, Ontario, Canada
[3] School of Dentistry, University of Otago, Dunedin, New Zealand
[4] Sir John Walsh Research Institute, University of Otago, Dunedin, Otago, New Zealand
[5] Dunedin School of Medicine, University of Otago, Dunedin, New Zealand
[6] Oral Health, AUT, Auckland, New Zealand
[7] Division of Urology, Department of Surgery, University of Western Ontario, London, Ontario, Canada
[8] Department of Biochemistry, University of Western Ontario, London, Ontario, Canada

Corresponding author
Simon M. Stebbings,
simon.stebbings@otago.ac.nz

## ABSTRACT

**Background.** A loss of mucosal tolerance to the resident microbiome has been postulated in the aetiopathogenesis of spondyloarthritis, thus the purpose of these studies was to investigate microbial communities that colonise the oral cavity of patients with axial spondyloarthritis (AxSpA) and to compare these with microbial profiles of a matched healthy population.

**Methods.** Thirty-nine participants, 17 patients with AxSpA and 22 age and gender-matched disease-free controls were recruited to the study. For patients with AxSpA, disease activity was assessed using the Bath Ankylosing Spondylitis Disease Activity Index (BASDAI). All participants underwent a detailed dental examination to assess oral health, including the presence of periodontal disease assessed using probing pocket depth (PPD). Plaque samples were obtained and their bacterial populations were profiled using Ion Torrent sequencing of the V6 region of the 16S rRNA gene.

**Results.** Patients with AxSpA had active disease (BASDAI $4.1 \pm 2.1$ [mean $\pm$ SD]), and a significantly greater prevalence of periodontitis (PPD $\geq 4$ mm at $\geq 4$ sites) than controls. Bacterial communities did not differ between the two groups with multiple metrics of $\alpha$ and $\beta$ diversity considered. Analysis of operational taxonomic units (OTUs) and higher levels of taxonomic assignment did not provide strong evidence of any single taxa associated with AxSpA in the subgingival plaque.

**Discussion.** Although 16S rRNA gene sequencing did not identify specific bacterial profiles associated with AxSpA, there remains the potential for the microbiota to exert functional and metabolic influences in the oral cavity which could be involved in the pathogenesis of AxSpA.

## INTRODUCTION

The term spondyloarthritis (SpA) is used to describe a spectrum of diseases which share common clinical features and a common genetic predisposition. Ankylosing spondylitis (AS) is the archetypal clinical phenotype (*Khan*, *2002*). The strong familial association amongst close relatives with AS is linked with the allele HLA-B27 present in approximately 90% of individuals with AS (*Roberts et al.*, *2013*). In 2009 the Assessment of Spondy-loArthritis international Society (ASAS) proposed new nomenclature for SpA, based on classification criteria (*Rudwaleit et al.*, *2009*), which has changed the perspective of anky-losing spondylitis (AS) (*Van der Horst-Bruinsma*, *2013*). The axial form of SpA (AxSpA) is now subdivided in the (early) non-radiographic type (nr-axial SpA) and the radiographic type, which equates to AS (according to the older modified New York criteria (*Van der Linden, Valkenburg & Cats*, *1984*)). The advantage of the new nomenclature is that it encompasses patients with earlier and less severe disease, and reflects the spectrum of severity seen in clinical practice.

A role for intestinal inflammation in the aetiology of AxSpA is suggested by an association with inflammatory bowel disease (IBD), where 20% of patients develop features of SpA (*Mielants et al.*, *1995*). In order to investigate aetoiological mechanisms associated with genetic predisposition in AxSpA, transgenic murine models expressing human HLA-B27 have been derived. These animals develop inflammation of the peripheral joints and spine and also develop colitis (*Hammer et al.*, *1990*). It is thought that the microbiome plays a role in disease pathogenesis, since germfree HLA-B27 transgenic rats do not develop arthritis (*Taurog et al.*, *1994*). However, arthritis develops when commensal bacteria, such as *Bacteroides vulgatus*, are introduced into these germfree models (*Rath et al.*, *1996*).

While the gastrointestinal tract harbours the largest microbiome in humans, other sites have microbial populations which are believed to significantly influence human health; in particular, the oral cavity harbours over 700 bacterial species. Although dental plaque represents a relatively small biomass, it is characterized by a highly dense microbial community which is almost as diverse as those found in the intestinal tract (*Costello et al.*, *2013*; *HMPConsortium*, *2012*). Additionally, when periodontal inflammation is evident, the junctional epithelium has greater permeability, with resultant challenge to microbial tolerance and immunogenicity (*Ohlrich, Cullinan & Seymour*, *2009*).

Periodontal disease is characterised by inflammation of the gingival tissues of the teeth and progressive loss of alveolar bone, resulting in apical migration of the epithelial attachment forming a periodontal pocket between the tooth root and the gingiva, thus creating an anaerobic environment (*Pihlstrom, Michalowicz & Johnson*, *2005*). Epidemiologic studies show that as many as 10% to 15% of the adult population have severe or advanced periodontitis (*Petersen & Ogawa*, *2005*). The oral microbiota has a role in periodontal disease and, in turn, is thought to play a contributory role in chronic systemic inflammatory diseases (*Pizzo et al.*, *2010*). For instance, patients with rheumatoid arthritis (RA) have a higher prevalence of severe periodontitis and tooth loss than healthy controls (*De Pablo, Dietrich & McAlindon*, *2008*) and the severity of periodontal disease correlates positively with disease activity in RA (*Ribeiro, Leão & Novaes*, *2005*; *Loyola-Rodriguez et al.*, *2010*).

The composition of the oral microbiota also appears to be a factor in RA (*Yeoh et al.*, *2013*) and treatment of periodontitis ameliorates RA disease activity (*Ortiz et al.*, *2009*). It is noteworthy that periodontal pathogens have been detected in the synovial fluid of patients with RA and AxSpA (*Kempsell et al.*, *2000*; *Martinez Martinez et al.*, *2009*; *Moen et al.*, *2005*; *Moen et al.*, *2006*).

We hypothesize that the microbiota present in dental plaque, which lies in close association with inflamed gingival tissue, allows bacterial components to trigger or perpetuate the inflammatory process in a genetically susceptible host with AxSpA. In order to test this hypothesis, we aimed to investigate the diversity of the microbiome in patients with AxSpA and compare this with healthy matched controls. In addition, we aimed to discover whether particular components of the oral microbiota were deferentially abundant in patients with AxSpA than in healthy individuals, to ascertain whether individual bacteria could be implicated in the pathogenesis of AxSpA.

## MATERIALS & METHODS

### Participants and sample collection and design

The pilot study used a matched case-control design. The case group comprised 17 with axial SpA as defined by the Assessment in Spondyloarthritis International Society (ASAS) criteria for AxSpA (*Rudwaleit et al.*, *2010*). These criteria were chosen to provide a sample with a wider range of disease duration and severity than would have been obtained using the modified New York criteria for AS. Patients were randomly selected from those attending the Rheumatology Department at Dunedin Hospital, New Zealand. Participants in the control group were purposively selected from the Electoral Roll and invited by post to participate in the study. They were individually matched to AxSpA patients on the characteristics of age (within two years), gender and ethnicity. In total 317 potential participants were identified from electoral roll. A total of 108 replied to letters sent out and of these, 53 volunteered to participate. Of the 55 who did not participate, 43 declined and six were not eligible due to exclusion criteria. Exclusion criteria for healthy controls were as follows: unable to provide informed consent, pregnancy, history of malignancy, taking anticoagulants, previous history of periodontal treatment, edentulous, history of inflammatory arthritis (including AxSpA), osteoporosis, significant cardiovascular disease, diabetes mellitus or inflammatory bowel disease.

All participants gave written informed consent in accordance with the Declaration of Helsinki, and ethical approval for the study was obtained from the Lower South Regional Ethics Committee, New Zealand (ref: LRS/10/06/020).

In the AxSpA patient group, all participants attended for clinical examination. Disease activity was measured using the Bath Ankylosing Spondylitis (BAS) disease activity index (BASDAI) (*Garrett et al.*, *1994*). All patients with AxSpA were genotyped for HLA-B27 and had a blood test at the time of sampling to measure C-reactive protein (mg/L). Recent radiographs were reviewed. All participants were surveyed about their smoking status.

## Oral examination

All participants underwent a detailed clinical oral examination following the World Health Organization (WHO) manual of oral diseases assessment (*Kramer, Pindborg & Infirri*, *1980*). The principal examiner (PS) underwent calibration prior to clinical data collection with intra-examiner calibration (with ten volunteers not recruited to the study). Data collected included: probing pocket depth (PPD) and gingival recession, clinical attachment loss (CAL) and bleeding on probing (BOP) at six sites per tooth. Other clinical information recorded included assessment of plaque deposits, oral mucosal conditions, and dental caries status: including enumeration of decayed missing or filled teeth (DMFT). The clinical assessor was not formally blinded to the status of the participant (AxSpA or control), this was impractical as many of the patient group had very obvious clinical features of AxSpA. Periodontitis was defined in this study as PPD $\geq$ 4 mm at $\geq$4 sites. There is considerable variation in the definition of periodontitis and no international consensus (*Savage et al.*, *2009*). In this study a definition of periodontitis used for epidemiological studies and defined by the Centre for Disease Control was selected (*Eke et al.*, *2012*). Observed clinical differences between the two groups were tested for statistical significance in SPSS (Version 21), using (as appropriate) Mann–Whitney $U$-tests and Chi-square tests, with an alpha value of 0.05.

## Plaque sampling and DNA extraction

Plaque accumulation was assessed to determine the oral hygiene status of AxSpA patients and controls. A plaque score was recorded for the buccal, palatal and lingual surfaces of all teeth except the third molars (*Silness & Löe*, *1964*). Subgingival plaque was sampled from all interproximal sites and placed in phosphate-buffered saline for storage at −20 degrees prior to analysis. Plaque sampled from different sites was pooled into a single sample for each individual. Plaque samples were centrifuged for 5 min at 13,000× g and the supernatant removed. Bacterial DNA was extracted using the Purelink Genomic DNA kit (Invitrogen, Carlsbad, CA, USA) according to the manufacturer's protocol for Gram-positive bacteria.

## Microbiota analyses

PCR of the V6 region of the 16S rRNA gene, sequencing and demultiplexing of sequencing reads was carried out as before using the Ion Torrent platform (*Bisanz et al.*, *2014a*; *Bisanz et al.*, *2014b*). Briefly, The primers used were CCATCTCATCCCTGCGTGTCTCC-GACTCAGXXXXXCWACGCGARGAACCTTACC and CCTCTCTATGGGCAGTCGGT-GATACRACACGAGCTGACGAC with XXXXX being a sample specific index. 25 cycles of PCR were carried out using GoTaq Colorless Master Mix (Promega) and amplicons were confirmed by gel electrophoresis. Amplicons were quantified and pooled at equimolar concentration using Qubit HSDNA (Life Technologies) before clean up (QIAquick PCR Purification kit). The resulting pool was sequenced at the London Regional Genomics Centre according to standard protocols using an Ion Torrent 316 chip. Raw reads are available in the NCBI Short Read Archive (accession: SRX513907) and the pipeline for demultiplexing and processing is available at github.com/ggloor/miseq_bin. Reads were filtered to only those containing perfect primer sequences, barcodes and a length between 70

and 90 for the V6 region. Taxonomy was assigned to operational taxonomic units (OTUs; 97% clustering) using the RDP classifier. $\alpha$ and $\beta$-diversity analysis and subsequent OTU level associations were made using the VEGAN and GUniFrac R Packages (*Chen et al.*, *2012*; *Oksanen et al.*, *2016*). To control for uneven sampling depth, all samples were subsampled to 9,064 reads. To examine differentially abundant taxa and to avoid spurious correlations common with the use of Pearson's correlation and proportional data (*Pearson*, *1896*), ALDEx2 (version 2.0.6) (*Fernandes et al.*, *2013*; *Fernandes et al.*, *2014*) was used to calculate the centered-log ratio (CLR) of each OTU within a sample from raw read counts. Additionally, this approach includes an estimation of technical variation common to microbiota datasets by randomly drawing Monte-Carlo instances from the Dirichlet distribution. Stringent multiple test correction was applied using Benjamini Hochberg false discovery rate. Associations were then examined in R software using Pearson's correlation coefficient. Differential abundance of taxa was represented in an effect plot (*Gloor, Macklaim & Fernandes*, *2015*). To completely recapitulate analysis, the raw code for read processing and analysis, associated metadata, and OTU table (.biom format) are available at github.com/jbisanz/ankspon.

## RESULTS

### Clinical and demographic characteristics of patients and controls

Demographic and disease characteristics of the 17 patients with AxSpA are presented in Table 1. Of the 17 patients, 14 fulfilled the older modified New York Criteria for ankylosing spondylitis (AS). Four patients with AS had additional SpA spectrum associated diseases including: two with psoriasis and two with inflammatory bowel disease. No patients had a history of reactive arthritis. The mean age of patients was 38 years and of controls was 37 years. Amongst AxSpA patients, disease activity measured by the BASDAI disease activity score showed a mean of 4.0, indicating a group with active disease; although a wide range of disease activity was recorded, with scores ranging from 0.2 to 8.9. Summary data on the two groups' oral disease characteristics are presented in Table 2. Patients with AxSpA were significantly more likely than controls to have clinically relevant periodontal disease (PPD $\geq 4$ mm at $\geq 4$ sites), more bleeding on probing and a higher mean plaque index.

### The plaque microbiota in AxSpA patients and controls

16S rRNA community profiling of the plaque samples yielded high sequencing depth, with the mean number of quality-filtered reads per sample being 51,203 (range 10,041–82,541, SD 13,549). Clustering was carried out at 97% identity to yield 127 distinct OTUs at $\geq 1\%$ abundance in at least one sample with an average length of 75 bp after removal of primer and bar code sequences. The OTUs were summarized to the family level and are displayed in Fig. 1. At the family level, the most abundant families across all samples were the Prevotellaceae (13.6%), Streptococcaceae (12.3%), Veillonellaceae (12.1%), Corynebacteriaceae (8.0%) and Actinomycetaceae (6.1%). At the phylum level, the most dominant phyla across all samples were the Firmicutes (28.2%), Bacteroidetes (22.5%) and Actinobacteria (19.3%), with the TM7 phylum being present at 0.3%.

**Table 1  Characteristics of AxSpA participants and healthy controls.**

| Variable (number unless otherwise stated) | AxSpA patients (n = 17) | Healthy controls (n = 22) |
|---|---|---|
| Female | 6 (35.3) | 7 (31.8) |
| Mean age (years; SD) | 38.0 (12.8) | 37.0 (12.7) |
| General health | | |
|     Cardiovascular disease | 11 (64.0) | 0 |
|     Diabetes | 1 (2.4) | 0 |
|     Inflammatory bowel disease | 4 (9.8) | 0 |
| Smoking status | | |
|     Never smoker | 14 (82.4) | 13 (59.1) |
|     Ex-smoker | 3 (17.6) | 6 (27.3) |
|     Current smoker | 0 (0.0) | 3 (13.6) |
| Number HLA B27 positive | 17 (100.0) | – |
| Evidence of sacroiliitis on imaging | | |
|     Radiographic | 14 (76.5) | – |
|     MRI | 2 (17.6) | – |
|     None apparent on imaging | 1 (5.8) | – |
| Mean BASDAI (SD) | 4.0 (2.1) | – |
| Mean CRP (mg/L; SD) | 5.4 (5.3) | – |

**Notes.**
Brackets contain percentages unless otherwise specified.
BASDAI, Bath Ankylosing Spondylitis Disease Activity Index; CRP, C-reactive Protein.

**Table 2  Oral health characteristics of AxSpA patients and healthy controls.**

| Variable (number unless otherwise stated) | AxSpA patients (n = 22) | Healthy controls (n = 17) | p-value |
|---|---|---|---|
| Mean decayed missing or filled teeth | 13.1 (7.6) | 9.9 (7.6) | 0.210 |
| Number of individuals with ≥4 sites with PPD ≥ 4 mm (%) | 11 (64.7) | 5 (22.7) | 0.008 |
| Mean number of sites with PPD ≥ 4 mm | 5.6 (5.1) | 5.1 (11.5) | 0.146 |
| Mean number of sites with CAL ≥ 4 mm | 10.1 (13.4) | 12.6 (25.1) | 0.318 |
| Mean number of sites with bleeding on probing | 36.6 (21.3) | 23.6 (17.6) | 0.041 |
| Mean plaque index score | 1.0 (0.4) | 0.6 (0.4) | 0.006 |
| Number meeting CDC severe periodontitis definition (%) | 1 (0.1) | 2 (0.1) | 0.709 |

**Notes.**
Brackets contain standard deviations unless otherwise indicated.
PPD, Pocket Probing Depth; CAL, Clinical attachment loss.

To address how microbial communities may differ between AxSpA and healthy controls, we applied multiple metrics of both $\alpha$ and $\beta$ diversity (Fig. 2). No alpha diversity metric showed a difference in richness or evenness of communities (Figs. 2A–2C) and no beta diversity metric showed clear clustering by community composition by principal coordinates analysis, which was statistically corroborated by ANOSIM analysis ($p > 0.05$).

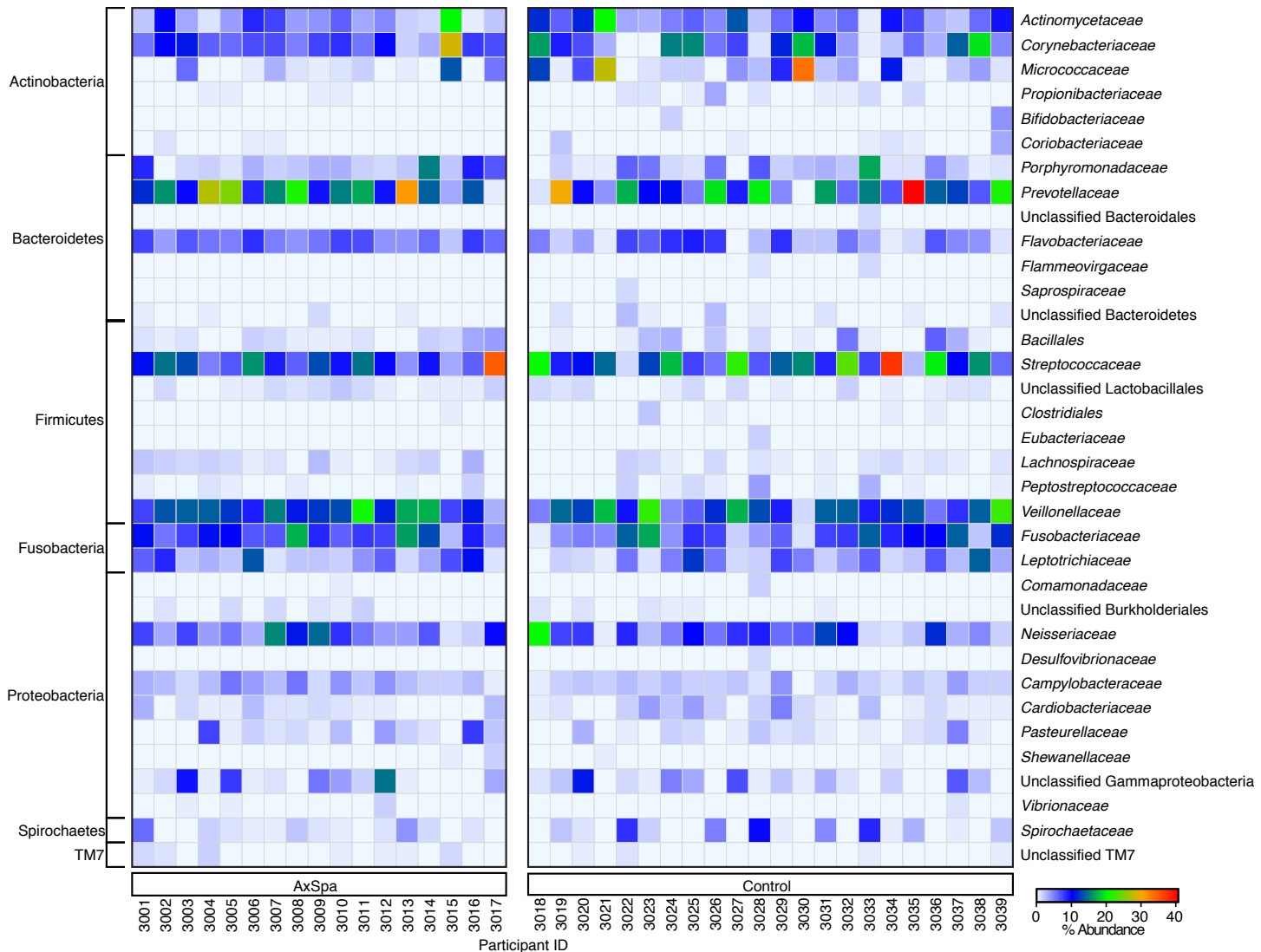

**Figure 1** Heatmap of family-level abundances (calculated post-OTU filtering) of the subgingival plaque microbiome in AxSpa and healthy controls.

Given that the microbiota in AxSpA patients and controls does not appear to differ at the community level, the analysis focused on individual OTUs and higher level taxonomic assignments (from phylum to species level) associated with both AxSpA and disease activity (BASDAI) in these individuals. No OTUs were associated with AxSpA, based on the robust approach applied with a stringent false discovery rate cutoff. Features were plotted in the context of an effect plot (*Gloor, Macklaim & Fernandes*, *2015*) demonstrating minimal difference between conditions (Fig. 3). The Actinobacteria phylum (unadjusted $p = 0.04$) was found to be higher in relative abundance in healthy controls, however this was not significant after multiple testing correction.

Next, in order to investigate the role of the microbiome constituents in disease activity, analysis focused on associations between OTUs (CLR-normalized) and disease activity

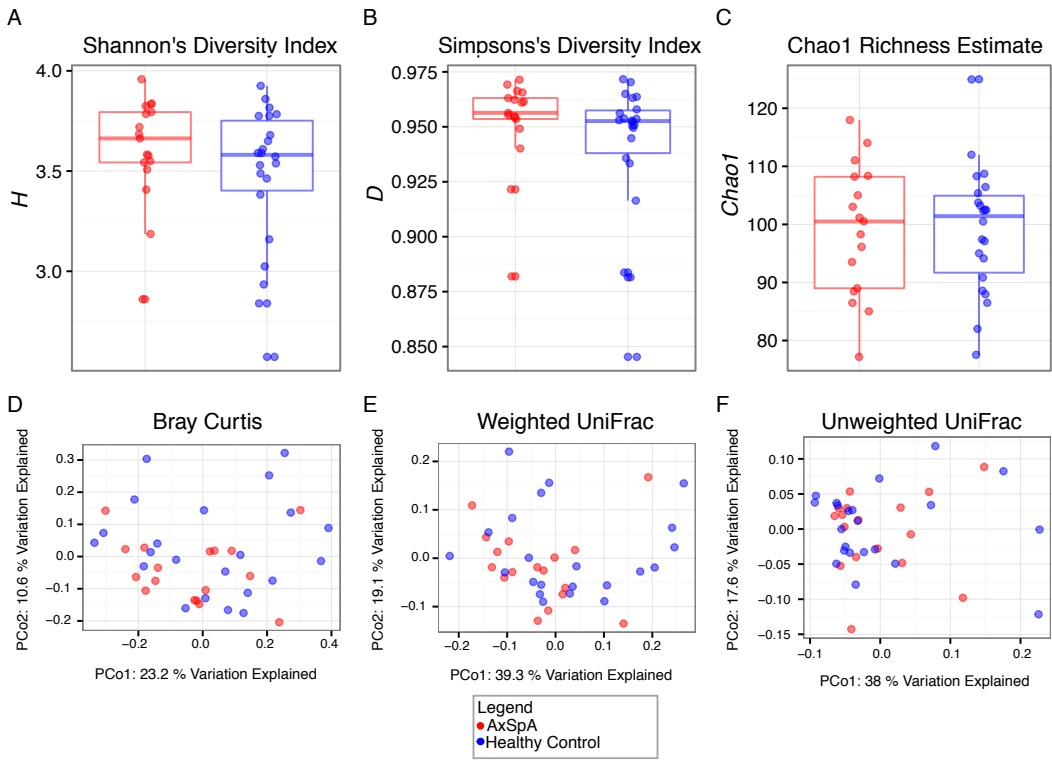

**Figure 2** **α and β diversity analysis of differences between AxSpa and healthy control communities.** Three measures of α diversity (Shannon's (A), Simpson's (B), and Chao1 (C)) all fail to report a significant difference between disease states in terms of organism richness and evenness (Wilcox rank sum $p >$ 0.05). Similarly, three measures of β diversity (both phylogenetic (E, F)/non-phylogenetic (D) and with (D, E) and without abundance weighting (F)) do not visually or significantly (ANOSIM $p > 0.05$) report differences between communities.

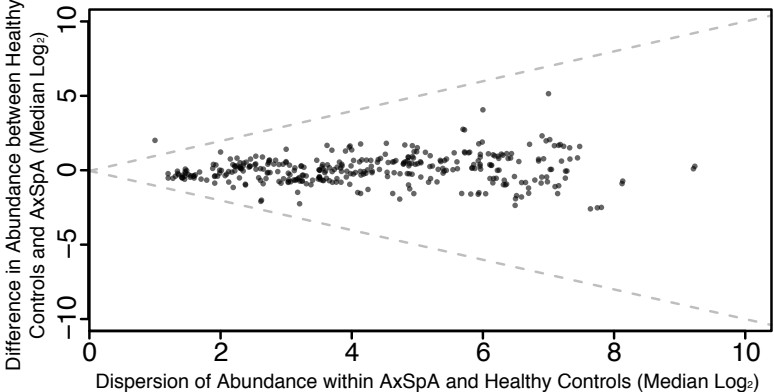

**Figure 3** **Differential abundance of features (OTUs and higher-level taxonomic assignments) shows a single feature (Actinobacteria; unadjusted $p = 0.04$) is higher in healthy controls, however this is not significant after multiple testing correction.** Each point represents a feature (taxa) while the $Y$-axis denotes the difference between healthy controls and AxSpA and the $X$-axis denotes the dispersion within groups. The broken lines demonstrate the case where difference between groups is equal to the dispersion within groups, and thus significant results would be expected to be found outside of the central area of the plot.

**Table 3  Correlation analysis of CLR-normalized OTU abundances with the oral health (PPD ≥ 4 mm at ≥4 sites).**

| OTU # | FDR | $r^2$ | Taxonomic assignment |
|---|---|---|---|
| 13 | 0.007 | −0.619 | *Streptococcus spp.* |
| 150 | 0.007 | 0.619 | *Saprospira spp.* |
| 0 | 0.014 | −0.586 | *Actinomyces spp.* |
| 21 | 0.08 | −0.522 | *Actinomyces massiliensis* |

(BASDAI), quality of life (ASQoL), and systemic inflammation (C-reactive protein) among the individuals with AxSpA. No associations were found with levels of C-reactive protein or BASDAI. Significant correlations ($p > 0.05$, $r < 0.5$) were found between 4 taxa and ASQoL; however, these were strongly influenced by a single individual and thus of doubtful validity. When this individual was removed from the analysis, the differences were no longer statistically significant. OTU relative abundances (CLR-normalized) were correlated with periodontal disease (PPD ≥ 4 mm at ≥4 sites) and examined across pooled data from healthy control participants and those with AxSpA (Table 3). Significant associations were apparent (FDR < 0.1) with high abundance taxa including *Streptococcus spp.* and *Actinomyces spp*.

## DISCUSSION

To our knowledge, this is the first study to examine oral plaque bacterial communities in relation to oral health in patients with AxSpA using high-throughput DNA sequencing techniques. At the community level, we found no difference between AxSpA patients and healthy controls in either their community structure or in the diversity of organisms in the plaque bacterial communities analyzed. This is despite the confounders that patients with AxSpA had significantly worse oral health than age- and sex-matched controls, with more plaque and a higher prevalence of periodontitis, despite their young mean age. In a similar study examining the oral microbiome in patients with RA, no difference in community profile was found amongst early RA, chronic RA and healthy controls, but differences in the subgingival microbiota were noted in relation to periodontitis (*Scher et al.*, *2012*).

Analysis at the OTU level failed to demonstrate an association between AxSpA and several bacterial taxa with stringent multiple testing corrections. While this suggests that no single causative-agent in the subgingival plaque is involved in the aetiopathogenesis, it should be noted that, without very many more samples, the sheer number of OTUs compared would make it unlikely that small effect-sizes would be detected. This is a limitation of the current study, and the data presented here should be viewed as a discovery-based approach warranting replicating studies with greater enrollment in order to conclusively rule out false negatives. All aspects of the data have been made publicly available for those researchers wishing to use the data set for hypothesis generation. The use of CLR-normalized relative abundances and modeling of technical error reduces the risk of false associations with low-abundance taxa and spurious correlations frequently reported in microbiota research (*Fernandes et al.*, *2013*; *Fernandes et al.*, *2014*). While being a relatively

new technology, 16S rRNA variable region surveys via high-throughput sequencing are a relatively coarse method of analysis, and further studies using shotgun-metagenomic and meta-RNA sequencing may provide more in-depth analysis that could discover strain or gene-level associations with disease. Furthermore, given the growing evidence for an association between periodontal disease and AxSpA (*Keller, Kang & Lin*, *2013*; *Pischon et al.*, *2010*), future studies should address the potential for the oral microbiota as a causative factor in AxSpA, as opposed to being simply a marker of periodontal disease and inflammation. This is particularly relevant given recent advances in our understanding of the genetic predisposition to AxSpA. Potential mechanisms can only be postulated. However, given evidence from elsewhere in the gut, it is possible that interactions between the autologous microbiome and the host immune system may be implicated in the aetiopathogenesis of AxSpA, since evidence for such a process in the ileo-colon continues to accumulate (*Costello et al.*, *2013*). At present, there is limited objective evidence that either specific bacteria or alterations in the microbiome as a whole influence the aetiopathogenesis of these conditions (*Yeoh et al.*, *2013*; *Costello et al.*, *2013*). Gingivitis and periodontitis are both host immune responses to plaque microbial challenge (*Seymour et al.*, *1988*). Credible biological mechanisms associating periodontitis and systemic inflammatory conditions have been proposed, but epidemiological confirmation of a causal relationship remains elusive. Associations are not necessarily causal and may comprise but one aspect of a variety of associated environmental risk factors (*Seymour et al.*, *2007*).

## CONCLUSIONS

Plaque samples from patients with AxSpA showed no difference in microbial diversity compared with matched controls, despite a higher prevalence of periodontitis. A wide variety of taxa were identified in plaque samples, many of which have been identified in previous studies both in the plaque of patients with periodontitis and in healthy individuals. These findings require further evaluation in larger samples in order to investigate bacterial community profiles and specific organisms of interest with reference to those identified in this study. Further, microbial profiles in the oral cavity should be investigated with other sites of interest including serum and synovial fluid specimens from patients with AxSpA.

## ACKNOWLEDGEMENTS

We are grateful to Mary Wallace for assistance with clinical data management.

### Funding

The W. Garfield Weston Foundation and New Zealand Dental Association Research Foundation provided funding for this work. JEB was the recipient of an NSERC Canada Graduate Scholarship. The funders had no role in study design, data collection and analysis, decision to publish, or preparation of the manuscript.

## Grant Disclosures

The following grant information was disclosed by the authors:

The W. Garfield Weston Foundation.

New Zealand Dental Association Research Foundation.

NSERC Canada Graduate Scholarship.

## Competing Interests

The authors declare there are no competing interests.

## Author Contributions

- Jordan E. Bisanz analyzed the data, wrote the paper, prepared figures and/or tables, reviewed drafts of the paper.
- Praema Suppiah and Nigel Yeoh performed the experiments, analyzed the data, reviewed drafts of the paper.
- W. Murray Thomson conceived and designed the experiments, analyzed the data, wrote the paper, prepared figures and/or tables, reviewed drafts of the paper.
- Trudy Milne conceived and designed the experiments, analyzed the data, wrote the paper, reviewed drafts of the paper.
- Anita Nolan conceived and designed the experiments, performed the experiments, analyzed the data, wrote the paper, reviewed drafts of the paper.
- Grace Ettinger wrote the paper, reviewed drafts of the paper.
- Gregor Reid reviewed drafts of the paper.
- Gregory B. Gloor contributed reagents/materials/analysis tools, reviewed drafts of the paper.
- Jeremy P. Burton and Mary P. Cullinan conceived and designed the experiments, contributed reagents/materials/analysis tools, wrote the paper, reviewed drafts of the paper.
- Simon M. Stebbings conceived and designed the experiments, contributed reagents/materials/analysis tools, wrote the paper, prepared figures and/or tables, reviewed drafts of the paper.

## Human Ethics

The following information was supplied relating to ethical approvals (i.e., approving body and any reference numbers):

Lower South Regional Ethics Committee of New Zealand (ref: LRS/10/06/020).

## DNA Deposition

The following information was supplied regarding the deposition of DNA sequences:

NCBI Short Read Archive: SRX513907.

## Data Availability

GitHub: https://github.com/jbisanz/ankspon.

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
