# Peer review of "The oral microbiome of patients with axial spondyloarthritis compared to healthy individuals"

_PeerJ, doi:10.7717/peerj.2095_

## Round 0.1 · original submission · Minor Revisions

I thank both reviewers for their input. They are both quite positive about your submission, however, as you will see down below, they have minor concerns that need to be addressed. One of the suggestions made by both reviewers is the need for clarity in the introduction. Please make sure the introduction is clear, and provides a more comprehensive background.

I also think it is important to acknowledge the limitations of the data and the approach used in this study. V6 is quite a short region, and analyzing microbial communities via taxonomy is not the most sensitive way to identify potentially differentially distributed, yet closely related populations across groups. I personally do not think your conclusions would have changed dramatically if you had used primers for longer amplicons, or relied on more sensitive approaches, however, it is always safe to acknowledge what you did not do, and that therefore some findings may have been missed.

Thank you for making your data available on SRA, and releasing the source code of your pipeline. All looks good. Unfortunately it was quite hard to browse the source code of individual scripts --if you would like to continue developing this pipeline, I would strongly suggest a complete re-organization of the codebase for a more testable and reliable collection of scripts as a future goal. I also would like to suggest the addition of VSEARCH in your workflow as an open-source alternative for search and clustering.

Please address each point raised by both reviewers in your response. I will be looking forward to reading your resubmission.

·

Basic reporting

In this paper, Bisanz et al. compare the oral microbiome of ankylosing spondylitis patients to that in age-matched healthy controls, using Ion Torrent sequencing of 16S rRNA gene amplicons. Using several alpha and beta diversity measures, no statistically significant differences could be found between the two subject groups.
Despite this negative outcome, this is a paper that is relevant to both dentists as well as clinicians.
Most figures were well presented, the text was well written, and analyses were generally well executed. My main comments would be to add more background in the Introduction, be more consistent with terminology, add information about the health status of controls, and add smoking and/or periodontitis as a possible confounding factors to the Discussion.

General comments
1. It would be great if the Introduction could include a bit of background about ankylosing spondylitis and axial spondyloarthritis. These terms are not familiar for the general audience or for microbiologists or dentists. Specifically, the title and some of the figure mention “ankylosing spondylitis”, but the rest of the paper uses the term “spondyloarthritis”, and I was not sure if these two terms are the same. Similarly, the term “axial spondyloarthritis” should be defined here – is it a specific subgroup of SpA? Then, the relationship between SpA/AxSpA and RA was also not clear. Is one a subgroup of the other? Could the authors add some sentences in the beginning of the Introduction to clarify these terms?
2. Related to the previous point, the 17 patients are defined by different terms. The paper switches from “AxSpA” (e.g. Abstract), to SpA (L83), and “Ankylosing Spondylitis” (Figure 2). AxSpA is used the most often, but it would help if this definition was kept throughout the paper, including Figure 2.
3. Figure 1 appears to show subject initials, e.g. KMP. Even if that is not the case, it would probably be better to stick to numbers (the numbers used with initials removed, or e.g. A1-17 and C18-39).

Experimental design

The experiment appears well designed, but it was not clear how healthy the "healthy controls" were.

4. It’s not clear whether the healthy control group consisted of really healthy subjects. Were the healthy controls also screened with the ASAS criteria? Since they were selected by electorate rolls, they were not necessarily healthy. L91 states they were excluded if they had teeth problems or diabetes, but there is no mentioning of their joint health or other health markers (these values are not given in Table 1). How certain are the authors that these controls were not affected by joint or bone diseases , inflammatory diseases such as Crohn’s Disease, or any other condition that might have affected their general health?

Validity of the findings

The findings were robust, but I have some some suggestions for (easy) additional analysis or discussion points.

5. The two subject groups appear to differ in prevalence of smoking. Several recent papers have found a significant effect of smoking on the oral microbiome (e.g. doi:10.1038/ismej.2016.37 and doi:10.1038/srep23745). Would recoloring the beta diversity plots according to current smoking status reveal any patterns? The authors don’t have to show that in the paper if there isn’t (but maybe make it a supplemental figure) but it might be an interesting small, additional analysis to add value to the paper.
6. Similarly, would recoloring the beta diversity plots according to periodontitis status reveal any patterns?
7. I did not understand the “differential abundances of features” analysis shown in Figure 4. Could the authors provide a bit more background in the Methods section?
8. This might be very personal (and this remark can be ignored), but it would make more sense to have the controls on the left and AxSpA on the right in tables and figures.

Additional comments

Specific comments
9. L47. Would the term “allele” be more exact in describing the HLA-B27 variant?
10. L107. What is meant by “It was not practical...”? Could the authors elaborate a bit what is meant here?
11. L113-115. Were all plaque samples pooled per subject? This is not clear from the text.
12. L120-121. Which software package was used for the quality filtering and demultiplexing? Were these the in-house scripts referred to by Bisanz 2014 PLOS ONE? It would be nice if some basic methods (e.g. PCR cycle number, quality filtering settings) were provided here.
13. L120-121. Were extraction controls included in the sequencing run, and did they yield any reads?
14. L138. Table 1 also includes the data for the Healthy Controls, not just data for the 17 AsSpA patients. Despite the title of this section, the first part of this section does not include a description of the controls.
15. L148-149. If the minimum number of reads per sample was 10,041, why did the authors rarefy to 9064 reads (L125)?
16. L150. Typo: “yeild”
17. L168. “altered in relative abundance” – in which direction? From Figure 1 it appeared more abundant in controls, but I was not sure.
18. Figure 1. Does this figure show the data for all OTUs or only the ones that remained after applying the criteria mentioned in L150/151? Also, the participant IDs should be made more anonymous (see remark above).
19. Figure 2. It would be more clear if there was more white space between the top panels and the names of the beta-diversity measures (e.g. “Bray Curtis”). Those names currently appear to belong to the top panels.

·

Basic reporting

Generally good. However, the Introduction, while linguistically good, is difficult to follow and contains information whose relevance to the study is not made clear. The Introduction should be clarified and sharpened.

Experimental design

The hypothesis stated in the fourth paragraph of the Introduction is not clear. Is the hypothesis that the presence of the microbiota in general can trigger or perpetuate inflammation in genetically susceptible hosts, or that particular as-yet unidentified members of the microbiota trigger or perpetuate inflammation in spondyloarthritis? The stated purpose of the study: “to investigate the oral cavity”, is also vague. The hypothesis and purpose should be stated more clearly.

Validity of the findings

Generally good. However, the results (lines 144-146), discussion (lines 185-186), and conclusions (line 220) all state that the AxSpA subjects had worse oral health than the controls, whereas two of the seven measures in Table 2 suggest that the controls had worse oral health than the AxSpA subjects (number of sites with CAL > 4mm and number meeting CDC severe periodontitis definition) and two of the remaining measures had non-significant p values. Please clarify why some measures are more meaningful or modify the claim of worse oral health.

Additional comments

For the aim of identifying within the microbiota particular species or strains that may be associated with spondyloarthritis, important information may be lost by clustering the data into 97% OTUs. The authors may wish to use an alternative method such as oligotyping to detect closely-related organisms with distinct distributions (e.g., Eren et al. 2014 PNAS 111(28): E2875-2884). Given their interest in the microbiome as it relates to periodontitis, they may also wish to identify which if any of their OTUs represent taxa associated with periodontitis such as Treponema spp., Porphyromonas gingivalis, and Tannerella forsythia.

---

## Round 0.2 · accepted · Accept

Thank you very much for the new version of the manuscript, and your detailed response to the reviewer comments. This clears the way for publication.

I have a very small comment: I find the title a bit hard to read (I tried to get used to it by reading it multiple times, but it did not help). If the authors feel comfortable with it, it is absolutely fine, but I wanted to point out my experience with it in case they would like to consider changing it slightly. In my opinion there are many ways to improve it. For instance, I thought this could be an easier-to-read, a bit more specific, and yet a much more dynamic alternative to the current one: "The oral microbiomes of axial spondyloarthritis patients compared to healthy individuals: A preliminary investigation with 16S rRNA gene amplicons". The authors should feel free to discard this suggestion if they are happy with their title.

Thank you again.